# The causes of loneliness: The perspective of young adults in London's most deprived areas

**Sam Fardghassemi** [ID]*, **Hélène Joffe**

Division of Psychology and Language Sciences, University College London, London, United Kingdom

* s.fardghassemi.17@ucl.ac.uk

## Abstract

Young adults are currently the loneliest demographic in the UK and other Western countries, yet little is known about how they see the causes of their loneliness. Thus, the objective of this study is to explore the subjective causes of loneliness among young adults (18–24 years old), particularly those of lower socio-economic status (SES) who are in employment, renting and living in the most deprived areas, since they are the loneliest in the UK. Utilising a free association technique and thematic analysis, and embedded in a phenomenological framework, the subjective causes of loneliness in a matched sample of 48 young adults in the four most deprived boroughs of London are found to cluster around five themes: The Feeling of Being Disconnected, Contemporary Culture, Pressure, Social Comparison and Transitions Between Life Stages. Disconnection arises from feeling one does not matter, is not understood or is unable to express oneself. Challenges pertaining to social media and materialism in contemporary culture contribute to loneliness as does pressure associated with work, fitting in and social comparison. Social media play a major role in exacerbating these experiences. Finally, transitions between life stages such as breakups, loss of significant others and transitory stages to do with education and employment are felt to cause loneliness. The findings suggest potential avenues for loneliness reduction.

## Introduction

Loneliness has become a major contemporary public health concern. Most vulnerable to loneliness are young adults aged 16–24 in the UK and other developed countries [1–6]. This was the case prior to the exacerbation of loneliness that the Covid-19 pandemic has created [7].

The experience of loneliness has detrimental effects on physical and mental health. Among young adults and adolescents, it is linked with immune deficiency, poor sleep, psychological stress, depression, and anxiety, among other conditions [8–13]. Given that 40% of young adults globally feel lonely often or very often [1] and the harmful effects that ensue, it is important to explore what young adults feel causes it in the hope of contributing to harm reduction.

A number of theoretical models may explain the causes of loneliness in youth including the social needs model, the cognitive discrepancy model, the multidimensional theory of loneliness and the evolutionary model. According to the social needs model, the desire for social connection is a basic, universal human need [14–16]. When this need is not met or satisfied,

**Data Availability Statement:** The Data are now available publicly. DOI number: 10.5522/04/17212991.

**Funding:** SF GM 156425 Grand Challenges UCL Grand Challenges Environment and Wellbeing

initiative https://www.ucl.ac.uk/grand-challenges/ This work was supported by a grant from the UCL Grand Challenges Environment and Wellbeing initiative (156425). The funders had no role in study design, data collection and analysis, decision to publish, or preparation of the manuscript.

**Competing interests:** The authors have declared that no competing interests exist.

one is likely to experience loneliness. Thus, loneliness can stem from a lack of social contact. In contrast, the cognitive discrepancy model proposes that one can feel lonely when there is a discrepancy between one's actual and ideal level of social relationships [17, 18]. This suggests that a person could have social connections, but these are less satisfying than they desire.

The multidimensional theory of loneliness distinguishes between 'social' and 'emotional' loneliness [19]. While social loneliness arises when one feels a lack of a social network (i.e., friends, colleagues, and neighbours), emotional loneliness can be experienced when intimate relationships or close bonds are deficient (i.e., a partner or close friend). Thus, one can feel emotionally lonely even if one has considerable social bonds. Similarly, one can experience social loneliness despite having intimate relationships. Weiss regarded the emotional aspect to be a more severely painful type of loneliness because one feels the absence of one's connections that are close, meaningful, and more emotionally bound. Weiss' theory of multidimensional loneliness is well supported empirically [20].

The way young adults approach social connections may determine the type of loneliness they experience. When the association between age, loneliness, and social engagement among adults of different age groups in the UK is examined young adults are found to value the quantity of relationships while older people value the quality [21]. This suggests that younger adults may be more focused upon increasing their friendship networks to protect themselves from social loneliness whereas older adults seek marital and family relationships to buffer against emotional loneliness. This corroborates socioemotional selectivity theory, which posits that as people get older and time is perceived to be more limited, they invest greater resources in emotionally meaningful goals such as spending more time with familiar individuals with whom they have rewarding relationships [22–24], as opposed to non-familiar people.

While young adults aim to increase their social networks to alleviate social loneliness, they may neglect their emotional needs and, therefore, be at risk of experiencing emotional loneliness. In fact, one study examining developmental trends regarding loneliness found that while emotional loneliness intensified across the adolescent and young adulthood years, social loneliness lessened [25]. Hence, emotional, and social loneliness may show distinct developmental trends across adolescence and young adulthood given the emphasis upon increasing social network, and possibly feeling a greater emotional distance to meaningful people in one's lives such as parents.

With the increasing use of social media in contemporary culture and focus upon growing one's virtual friendship network or the number of "friends" or "followers" one may have on social technologies, young adults may be at risk of lacking depth and meaning in the quality of their online friendships and ultimately feeling emotionally lonely.

While the aforementioned theories see loneliness as a negative experience, the evolutionary model [26], which also proposes that human beings have a universal need to belong, puts forward that loneliness indicates essential social relationships are lacking, under threat or regarded as poor in quality [27] and motivates people to make an effort to improve their social bonds. Consequently, loneliness lessens, and people start to experience a greater sense of well-being from their social connections with others since relating to others is a basic psychological need. This regulatory process of rebuilding or enhancing one's social bonds when they are experienced as deficient, has helped human genes to survive and continue to reproduce [28].

Another cause of loneliness among young adults is linked with educational, employment and household transitions. The highest proportion of British young adults who reported feeling "often or always" lonely was found to be among 18- and 21-year-olds who had left home and transitioned into university, and then transitioned into employment [29]. A second peak in loneliness was found among the oldest students in a study of 18–20-year-olds, which may be related to transition into full-time employment and adapting into early adult life [30]. In

addition to these transitions, there are also relationship transitions such as those with a romantic partner [31, 32].

Social media are also a possible cause of loneliness in young adults, though the evidence concerning the association between its use and loneliness is equivocal. There is continued debate between the displacement and stimulation hypotheses on the association between social media use and loneliness e.g., see [33, 34]: Some researchers support the hypothesis that social media use increases loneliness because it displaces offline face-to-face interactions [34, 35]. Others stand by the hypothesis that social media reduce loneliness because it offers the opportunity to stimulate existing relationships and form new ones [e.g., 33, 36]. Other studies, however, show that the validity of each hypothesis above depends on the level of analysis. In one study, on days when people used social media often, they were less likely to engage in in-person social interactions but, at the between-person analysis, those who spent a large portion of their time on social media did not engage less in face-to-face interactions than those who refrained from it [37]. Longitudinal studies, meta-analyses, mixed-methods, and qualitative research show either small, weak or no support for the association between the quantity of social media use and increases in loneliness among student samples and all age groups [38–42]. What these studies suggest is that the quality of social media use rather than quantity may better determine its impact on loneliness.

Social comparison on social media, in particular, is one engagement type that has received growing attention in recent years [43, 44]. Studies find that those who passively use social media (e.g., seeing other people's profiles but not directly interacting with them), in comparison to those who actively engage with them (e.g., by chatting via instant messaging or broadcasting in the form of posting a status update), are more likely to engage in social comparison behaviours and feel jealous towards others [45–47]. A meta-analysis corroborates this notion that while social media's passive use is negatively associated with loneliness, active use enhances wellbeing [48]. The social environments that teenagers operate in may influence social comparison tendencies on social media. In one study, competition among friends in a peer-group was found to be a strong predictor of orientation towards social comparison and feelings of envy in teenagers who were exposed to intensive social media use [49].

In addition, lower social-economic status may be a risk factor for loneliness in young adults. Measures of socioeconomic status, such as low family income and low social class based on parental occupation are each related to higher levels of loneliness among adolescents [50, 51]. These findings corroborate the findings by the UK's Office for National Statistics, which found socioeconomic status to be associated with loneliness in young adults [5]. Access to financial means may facilitate participating in activities that reduce loneliness.

## Current study

While previous research indicates potential causes and risk factors associated with loneliness in young adults, little is known about how they conceptualise the causes of their loneliness. In light of this gap and the major public health problem that levels of loneliness currently pose especially in young adults, this study sought to better understand young adults' sense of the cause of their loneliness. In addition to age, lower socioeconomic status, in employment, renting and living in the 50 percent most deprived areas were associated with higher levels of loneliness [5]. Therefore, the current study aimed to conduct an in-depth qualitative analysis of the subjective cause of loneliness among this demographic specifically. Since the sample identified with these characteristics and circumstances has recently shown to be a lonely demographic [38], it is imperative to understand the causes of their loneliness; this will then aid the design of targeted interventions to diminish loneliness in this group. Furthermore, since social media

have become a major part of young adults' social lives [52], and the literature contains contradictory findings regarding its impact on loneliness, the present study also set out to examine the subjective role of social media on young adults' sense of loneliness.[1]

This research is concerned with the study of an individual's lived experience. This makes it closely aligned to a phenomenological framework [53, 54]. It aims to explore the subjectivity of the individual but also the way people come to share similar understanding of the world, their common perceptions of it and the way they create a sphere of intersubjectivity. It further intends to uncover the meaning that people give to a phenomenon and gather rich description of how it is concretely lived [55]. Given that the theory concerns itself with the way the world appears to the person experiencing it [56], it provides a good framework for examining young adulthood, a period of exploration in light of its many transitions: household/relationships, education, and employment. Of course, how the world presents itself to young adults is shaped in large part by the social media and this is worthy of depth exploration. In addition, there are no studies of how lower socioeconomic status and loneliness may be interlinked (other than our earlier work, see [38]), beyond the lack of funds preventing participation in certain activities. Phenomenological exploration can offer insight into this link.

The following research questions will be addressed:

- What are the subjective causes of loneliness for young adults in London's most deprived boroughs?

- What role do social media play in the subjective cause of loneliness for young adults in London's most deprived boroughs?

## Materials and methods

### Neighbourhood selection

Participants were recruited from the London boroughs of Newham, Hackney, Tower Hamlets, and Barking & Dagenham. These boroughs are ranked as the most deprived in London by the Index of Multiple Deprivation (IMD). The total relative measure of deprivation for this index is calculated based on the combination of the following seven domains: income, employment, education, skills and training, health and disability, crime, barriers to housing services and living environment [57]. Since living in the most deprived broughs has been found to be a condition associated with higher levels of loneliness among young adults in the UK [5], the present study chose the selected broughs for an in-depth study.

### Participants

The researchers requested that a recruitment agency recruit a purposive sample of 48 respondents. Participants were selected based on the characteristics associated with greatest loneliness [5]: British born (23 males, 24 females and one 'other') young adults between the ages of 18–24 ($M = 21.23$, $SD = 2.43$), from lower socioeconomic backgrounds (C2DE)[2], in employment (full- or part-time) and renting in the most deprived boroughs of London: Newham (N = 12), Hackney (N = 12), Tower Hamlets (N = 12) and Barking & Dagenham (N = 12) [57]. All interviews were conducted between May and August 2019.[3] Further demographic characteristics are shown in Table 1.

### Procedure

The researcher handed participants the consent form, information sheet outlining the nature of the study (e.g., a research project exploring young adults' social lives in the city), the free

**Table 1. Young adults' (18–24 years old) demographics (in numbers and percentages)ᵃ.**

| Demographic categories | Male | Female | Other | Total |
|---|---|---|---|---|
| **Boroughs** | | | | |
| Newham | 6 | 5 | 1 | 12 (25%) |
| Hackney | 5 | 7 | - | 12 (25%) |
| Tower Hamlets | 7 | 5 | - | 12 (25%) |
| Barking & Dagenham | 5 | 7 | - | 12 (25%) |
| **Race** | | | | |
| Black, Asian & Minority Ethic (BAME) | 17 | 15 | 1 | 33 (68.75%) |
| White | 6 | 9 | - | 15 (31.25%) |
| **Religion** | | | | |
| Christian | 10 | 6 | 1 | 17 (37.5%) |
| Muslim | 9 | 5 | - | 14 (29.17%) |
| No religion | 5 | 7 | - | 12 (25%) |
| Other | 3 | 0 | - | 3 (6.25%) |
| Prefer not to say | 1 | 1 | - | 2 (4.17%) |
| **Total** | 23 | 24 | 1 | 48 |

a The questionnaire about religion presented participants with a list of options including: 'Jewish', 'Buddhist', 'Hindu', 'Sikh', 'Christian', 'Muslim', 'No religion', 'I'd rather not say', and 'Other (please specify) . . . . . . .'. In the table, we only included the religions or options that participants from the present study were affiliated with.

association tasks and a list of self-report questionnaires.4 Participants were kept blind to the specific aim of the study. Once they had given consent to be interviewed and audiotaped, they undertook the free association tasks, followed by the elaboration interviews. After the study they were debriefed as well as given details of professional services for further support in case the interview evoked emotional distress that required professional support. Participants also received a small cash incentive. This study obtained ethics from the researchers' University Research Ethics Committee (CEHP/2013/500).

**Free association task.** Free associations reveal people's subjective thoughts and feelings about an issue under investigation with minimum researcher-led interference. Thus, they elicit 'stored', naturalistic ways of thinking. A free association technique, termed the Grid Elaboration Method (GEM), was used. Its origin is within the psychoanalytic technique, which advocated that when people are given the opportunity to create their own stream of images and ideas, buried feelings and motivations that underpin their conscious-level behaviours will be revealed [58]. In line with this technique, a free association task was administered: young adults were presented with a piece of paper that contained a grid of four empty boxes and asked to write and/or draw what they associated with the experience of loneliness. They were further required to provide one image or idea per box5 (see Fig 1). The prompt for the free association was also piloted with two participants from the chosen boroughs before the study began and showed the 'the experience of loneliness' phrase was most effective.

**Semi-structured interviews.** Participants were required to elaborate on what they had written and/or drawn for each box in an interview that followed the free association tasks. This started with "can we talk about what you have put in box 1, please?" Once the participant had elaborated on the first free association, the same process was followed for the three remaining boxes in sequence. Prompts such as "can you tell me more about that?" were used to ensure that participants' thoughts and feelings about the experience of loneliness were fully explored and emerged naturalistically without insertion of content from researcher questioning. Each interview lasted for approximately 60 minutes.

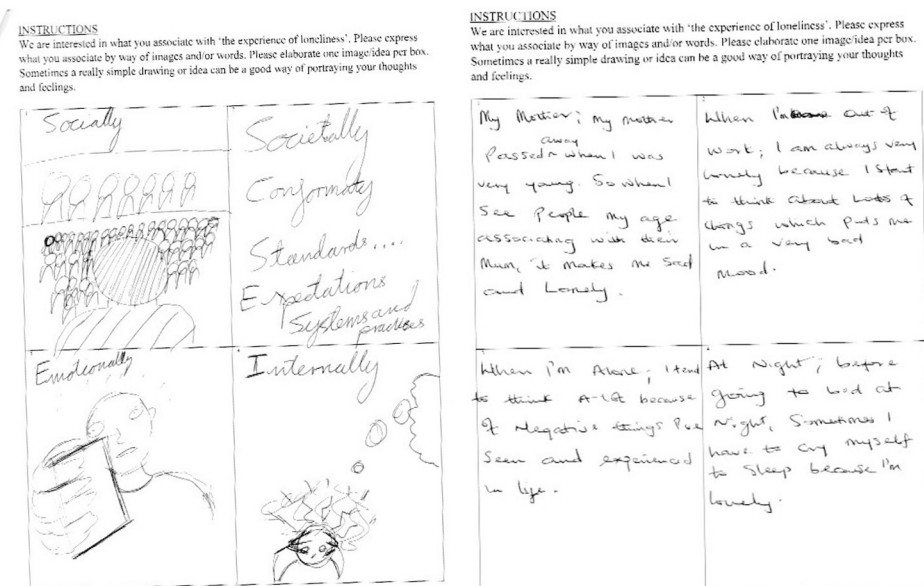

| Male, aged 19, Hackney | Female, aged 23, Barking & Dagenham |

**Fig 1. Examples of the completed free association grids.**

## Data analysis

A thematic analysis was conducted on the interviews in which participants elaborated their own grids [59, 60]. Transcripts were read carefully a number of times and key ideas contained in them noted and categorised. The creation of inductive codes ensured a naturalistic development of the codes from what was observed in the dataset. The codes were subsequently grouped into sets in a coding manual and key themes were identified. This allowed researchers to gain access to a more complex picture of young adults' lines of thinking and feeling beyond condensed initial free associations. To test the reliability of the coding frame, once the first coder had coded all of the interviews with the coding manual, he approached a second independent coder and trained him to use the coding manual. The second coder double-coded four of the interviews covering the demographic groups represented in the dataset on Atlas.ti 8. Inter-coder agreement analysis revealed an average Krippendorff's Cu-Alpha of 0.642 across all the codes, which indicates substantial reliability [61]. On this basis, the coding manual was refined (e.g., code names and descriptions were changed to create more transparency). There is a current debate about whether thematic analyses should go through reliability checks [62, 63], it provides systematic basis for the analysis. Subsequently, the full data set was analysed using Atlas.ti 8.

## Results

### Interview themes

The following sections will outline the most prevalent themes found in young adults' elaborations of their own free associations concerning the causes of loneliness (see Table 2 for theme summary) and then expand on each in turn. Responses were also compared across gender, race, and religion within each theme.6 Finally, a symbol used by young adults to metaphorically capture the theme is added at the end of each theme.

**Table 2. Subjective causes of loneliness among young adults in London's most deprived areas: Theme summary.**

| |
|---|
| 1. The feeling of being disconnected (65%)b |
| a) Not being able to express myself, my feelings, or my issues |
| b) Not feeling understood |
| c) Feeling I don't matter |
| 2. Contemporary culture (54%) |
| a) Social media as fake portrayals of reality |
| b) Changing one's identity/not being true to oneself |
| c) Lack of face-to-face social interactions and care from people |
| 3. Pressure (52%) |
| a) Pressure to fit in or be accepted |
| b) Pressure related to working, finding a career, earning a living and money |
| 4. Social comparison (48%) |
| a) Social comparison–general |
| b) Feeling stuck or behind while everyone else is progressing |
| 5. Transitions between life stages (48%) |
| a) Relationship breakups and losses |
| b) Transitory periods |

b The percentages in bracket represent the proportion of the sample who mentioned the theme.

**Theme 1: The feeling of being disconnected.** The most prevalent cause for loneliness was feeling disconnected. Many young adults spoke about being lonely because they felt unable to express themselves, their feelings or talk about their issues. They also talked about being lonely due to feeling they did not matter to others and were not understood.

a) *Not being able to express myself, my feelings or talk about my issues.* Young adults felt unable to express themselves, their feeling or talk about their issues for a number of reasons. The most common factor was the fear of being judged, rejected, or viewed in the 'wrong way' by others. Directly and indirectly, a number of young adults said the pressure of expectations, norms and how one is supposed to be as well as the pressure to fit in made them fearful of expressing themselves or talking about their issues:

*". . .I struggled. I sometimes, although when everyone was around me, I struggled to open up about me, I feel as though they would judge me and judge me in the way, what I said, is it socially acceptable, how can I do this, so sometimes I bottle things up and keep things to myself. . ."* [Female, aged 22, BAME, Tower Hamlets]

Participants stated that even if they were surrounded by other people such as friends and family, they could still experience loneliness if they felt unable to express themselves, their feelings, or their issues. Other factors that formed part of feeling disconnected included the sense that others did not understand one or what one was going through and being "different" (i.e., sexuality, lifestyle, physical appearance, body size, views). In particular, being from an LGBT + background was mentioned more than the other factors pertaining to being different as a cause of loneliness. This was mentioned by both participants who identified themselves as LGBT+ and those who knew someone from this background. This was accompanied by a fear of judgement, rejection, and isolation.

b) *Feeling I don't matter.* The feeling of not mattering was another cause of loneliness. This was spoken about mainly in terms of not having anyone to turn to or talk to, not being

heard, or listened to, not cared for and not supported. Not mattering occurred in numerous contexts such as when family or friends were not there for one, living alone and having no one to talk to, and needing support about education and career and not receiving it. For example, a female participant expresses not speaking with anyone makes her feel uncared for and lonely:

*". . .I put myself in- I'd put myself in like this situation. Like I thought about myself in a way. Like. . . so like. . . not speaking to someone about like how your life is going and like sometimes it can make you- sometimes it can make you think like does anyone even care about me? Like, do people even think- think of my name anymore? Like. . . like not talking about your worries or your ambitions can make you feel lonely. . ."* [Female, aged 18, White, Hackney]

c) *Feeling not understood.* The feeling of not being understood was also a common contributor to loneliness. In particular, participants spoke of the feeling that others do not understand what one is going through, what one is trying to achieve or why one is doing certain things. They also felt that they were the only ones going through a situation, particularly a difficult one and others did not understand their experience. For example, a female participant expresses going through a challenging time and feeling others around her are not going to relate to it:

". . .what I'm going through is so extreme and it's only me in this, and people around me are not even going to get it, so I'm just gonna like be consumed by what I'm going through, not reach out' and that leads to me just feeling even more lonely, lonelier than I did before, yeah."

[Female, aged 18, BAME, Newham]

Feeling misunderstood was further accompanied by sadness and depression, rumination, and negative thinking such as "maybe, I am the problem." Additionally, participants felt that others could not relate to them due to their sexuality, physical appearance, anxiety and differences in views and upbringing. Some also said that when they felt not understood they removed themselves either physically or emotionally.

The theme of being disconnected is symbolised as being in a sea with no one:

*"Say you got lost at sea, with you and a- a life raft, and you're on sea for three months, you'd be very lonely because you've got no one. You've got no one to talk to for them three months, it's just you and nothingness. So maybe. . . I associate sea with loneliness because that's what the sea is. The sea is probably the most loneliest place in the world."* [Male, aged 19, White, Barking & Dagenham]

**Theme 2: Contemporary culture.** Issues to do with contemporary culture were another cause of loneliness in over half of the sample. Such issues pertained to social media, distancing from one's true sense of self, and lack of face-to-face communication and care. Other concerns included competition and obsession with material gains in contemporary times. These are further elaborated below.

a) *Social media as fake portrayals of reality.* Social media were felt to portray reality in a 'fake' way and contributed to young adults' sense of loneliness. Participants said what they see on social media posted by others are only what people want them to see. For example, seeing

others on social media going on holidays, having money, buying luxury goods, and portraying a successful life were all just for "show" and did not give the "real story". People might actually be lonely but portray the opposite on social media. There were mentions of phrases such as "it is fake", "seeing the highlights and that is what fools you", "you don't get to see their true emotions", and "so distant from reality". In particular, a number of young adults also expressed that they, along with others, would never post their sad or lonely moments on social media. Everyone would share "happy pictures", specifically those that show their "best selves". Consequently, a significant portion of young adults felt that sometimes seeing photos and videos of others on social media even though they were fake portrayals of reality brought about feelings of loneliness, sadness, envy, jealousy, or self-doubt:

". . .when you go on Instagram and stuff like that, there can be a lot of like fake portrayals of reality and that can kind of distort the way you see yourself, your goal and things like that."

[Female, aged 18, BAME, Newham]

b) *Changing one's identity/not being true to oneself.* Similarly, changing one's identity or not being true to oneself was another cause of loneliness. Participants were under a lot of pressure to act or behave in a certain way or be somebody that they were not just so that they could be accepted or felt a sense of belonging, however, this conformity distanced them from their true identities and led to feeling lonely. Participants were also inclined to dissociate from their true selves due to the fear of being judged or rejected. Overall, social media played a role in creating an environment where young adults felt pressured to build an online persona that was not true to who they were on a day-to-day basis just so that they could fit in and be accepted. This experience contributed to feelings of loneliness. A young male participant expresses the notion that not being oneself results in loneliness:

*". . .When I'm around people, I have to be what people want to see which as I said conveys the, conveys a person that's not me, so in retrospect, if it's not me, I'm not getting those type of chemicals in my brain or those type of interactions that my character should be getting, I said that in a third person but it's like, it's just lonely because, you're not, it's not you, you're not you when you're out doing stuff, you're what people wanna see. . .".* [Male, aged 24, White, Barking & Dagenham]

c) *Lack of face-to-face social interaction and care from people.* A lack of face-to-face communication and empathy from people in contemporary times was also felt to contribute to loneliness. Participants lamented that everyone in today's era or in London specifically is busy working and living the high pace life of the city. Some said everyone is only looking out for themselves, focusing on achieving more, chasing after money, and competing with one another. As a result, there is no sense of community, sharing or trust. People have little time or empathy for others. One of the ways that these are manifest is that people have little time to have genuine interactions with strangers, check up on or greet one another with a basic "hello", "how are you?", "good morning" or even smile at others in public. In particular, trains and public transport were places where young adults felt most lonely because despite being surrounded by many people, little interaction or eye contact takes place. Commuters are busy using their phones instead of talking to others who are physically around them. A male participant expressed his experience of this notion through the following quote:

*". . . if you do live in London, you have a lot to deal with on a daily basis regardless of if you're having a quiet day, this couldn't be a lot of what's happened. And so, we're all caught up in our lives enough that we that becomes difficult to become even empathetic to people and so you do get a lot of time when you look around a train and [see an] empty face looking at you not even with any judgement or anything like that, that sort of thing adds to the whole the coldness of loneliness, I guess in that sense."* [Male, aged 24 White, Hackney]

Finally, contemporary culture as a theme is symbolised as a rat race:

*". . .that's what the world is, it's the rat race. Chasing money, chasing- chasing this world. The rat race. Put rats in a race to, you know, get the cheese. They dodge- forever try to chase the cheese, you know, and once the cheese finishes, they need to go get more. They're- they're just stuck in a race. It's never ended, is it?"* [Male, aged 23, BAME, Hackney]

**Theme 3: Pressure.** Pressure was another major cause of loneliness. Young adults either directly expressed their sense of pressure or indirectly indicated they were suffering from it. In particular, the pressure to fit in or be accepted and pressure regarding one's working life contributed to loneliness. Social media magnified young adults' experience of pressure.

a) *The pressure to fit in or be accepted.* The pressure associated with fitting in or being accepted was the most common of pressures experienced. The main contributor was the fear of being judged, excluded, rejected, isolated or left out by friends, social groups or society at large. Hence, young adults would change the way they acted, talked, and dressed even if they did not want to, just so that they could feel they belonged or were accepted. Some further explicitly expressed a sense of unhappiness as a result of having to alter themselves in order to fit in:

*"It almost makes me feel like I was faking everything. Like I was living this fake life. You know, I wasn't really myself because myself is very different to what I was portraying. I used- it's like- it made me very sad, the fact that, you know, I have to portray this- someone else just so- so that I can fit in with the society like. . . you know, I used to- almost like I- yeah, I'm living but it- I'm not living my life. I'm living someone else's life. And that if not often times would result in like extreme depression. . ."* [Male, aged 22, BAME, Tower Hamlets]

Young adults also voiced their concern for the pressure of expectations in regard to what they were supposed to be and how they were expected to act, dress, and look in order to be accepted. For example, one participant, in particular, said one "can't laugh wrong, smile wrong". Similarly, they felt pressured to appear good looking both offline and on social media in order present an accepted image. Some of the participants compared their physical appearance to friends or celebrities on social media and as a result either questioned their worth or felt insecure about how they looked. The pressure concerning physical appearance was accompanied by that for material gains in order to fit in. Participants stated that today's young adults are mainly concerned with buying material goods, mainly cars, shoes, sunglasses, and travel. Again, participants felt that social media amplified this experience because it prescribes, by way of its imagery, how to dress and what to own. There was also the pressure to keep up a façade that one is happy on social media when in reality one is not. There were mentions of either the participants themselves or somebody else who is sad in reality, but their social media profiles show them as "happy", "on a beach" or portray their "best selves". Additionally, there was pressure to be popular on social media because it was considered a popularity contest

since it is perceived to be about who has more 'followers' and 'likes'. However, participants viewed this negatively since large follower numbers did not mean one was cared for or actually liked.

b) *Pressure related to working, career, and money.* The pressure associated with working was another highly common experience that led many participants to feel lonely. In particular, they felt the pressure of working too much and this left them with little time for social activities, which they were not happy about:

*"I would rather be doing what I want to be doing, so I have more free time to see the people that I want and enjoy my life a bit, but when you're just constantly in work, it's like, it just feels like a routine control system and, you know, even not, hardly seeing family, as well, can lead you to being quite, like if you're always working, leads you to feeling quite lonely. . ."*
[Non-binary, aged 18, BAME, Newham]

There were also mentions of the pressure to have a good job in London and to perform well in one's role as well as to strive for achievements and success. Furthermore, the pressure of working was sometimes associated with a negative view of doing the same routine work and accompanied by worrying about money: whether one has money to eat, pay rent and meet with friends out and sometimes the lack of money was mentioned as a barrier to socially connect with others. Finally, "London" was viewed as a pressurised space and "today's society" as a pressurised era due to its constant working lifestyle.

Finally, pressure as a theme is captured symbolically by a male participant who felt the heaviness of having to conform to society's expectations:

*". . . because he's constrained himself just being, 'oh, I'm just a fish', he's just gonna be a fish. But then you know, why can't it be a flamboyant fish with legs and just walk out of that tank or just walk about that tank in a different way?"* [Male, aged 24, White, Hackney]

**Theme 4: Social comparison.** Almost half of the sample, and more so for females than males, saw social comparison as a cause of loneliness. Comparisons were made on a range of matters from achievements to physical appearance and friendships. There was also a sense of comparison in terms of feeling behind or not changing while everyone else is improving themselves. Such comparisons accompanied a range of negative feelings such as jealousy. Finally, social media magnified young adults' sense of social comparison.

a) *General social comparison.* Participants, particularly females, socially compared themselves with others in terms of a range of factors including physical appearance, jobs, salaries, family dynamics, being in a romantic relationship or married, knowledge, friendships, happiness, travel, and material possessions both offline and on social media. They also compared their own experience of being at home alone while seeing photos of friends out having fun on social media. Overall, these comparisons led participants to experience a range of negative emotions such as loneliness, jealousy, sadness, doubt about themselves, disappointment, low self-worth, and distortion in the way they saw themselves and their goals. For example, a female participant describes her experience of seeing others having a social life at work on social media whilst she felt she did not have one:

*". . .I used to see on their social media they were having a laugh, they used to get pizzas for lunchtime, and just little stuff like that, I felt as if why couldn't I fit in like that, why do I feel*

*like an outsider, what's different between me and them. . .?"* [Female, aged 22, BAME, Tower Hamlets]

Although these comparison tendencies were evident among the sample, they were more common across females from BAME backgrounds or females identified as having no religion.

b) *Feeling stuck or behind while everyone else is progressing.* Feeling trapped while everyone else is improving is also a cause of loneliness. Everyone is seen to be "progressing", "growing", "changing", "succeeding", "reaching their goals" and "improving" themselves while participants felt "stuck", "behind" or "not doing enough". In particular, there were mentions of everyone going to college, getting a job, marrying, buying a car, travelling and participants felt they were not achieving any of those. As a result, they expressed feelings of sadness, resentment, low self-worth, and inadequacy:

*"It makes me feel upset because I feel like I haven't changed like everyone around me is changing, moving away to better themselves and I feel like I'm still stuck in the same place that I haven't changed, and then maybe it's me that's like the problem. . ."* [Female, aged 20, White, Barking & Dagenham]

Finally, social comparison is symbolised as being in darkness while friends are under the sunlight:

*"So, I'm the one alone, there are two people on the right-hand side, they are basically my mate and his wife and the sun at the top is shining at them. So, I'm more towards the dark side, where I haven't improved my life, I haven't gone through what the phases that they have gone through. I haven't succeeded the way they have succeeded. So, the sun shining at them because they have the flashy cars, they got the designer goods, they are married, they been travelling abroad, they been to a lot of places. . ."* [Male, aged 23, BAME, Tower Hamlets]

**Theme 5: Transitions between life stages.** Nearly half of the participants also considered changes in life events such as relationship breakups or losses and transitionary periods such as moving from school to university or from university to working to be a cause of loneliness.

a) *Relationship breakups and losses.* Participants said romantic relationship breakups are a lonely experience because having got used to their partner and certain routines all of a sudden both disappear. In particular, the aftermath of break ups was mentioned as a lonely period. Hence, it can lead to feeling sad, angry, and depressed. For example, a male participant describes the experience of loneliness after a breakup to be unique due to a complete change in lifestyle:

*". . .everybody from all points of life can experience loneliness, but what particularly stands out to me is the type of loneliness one experiences after a breakup, where you go from one extreme to the next, you go from knowing everything about this person to com-, completely not knowing what this person is doing or whatever and then it's like the whole talking every day, every night, and then not talking at all, it's just complete cut off, so but that obviously that's it, that's an experience of loneliness, in my opinion. . ."* [Male, aged 23, White, Newham]

Participants also mentioned a number of ways one can overcome breakup loneliness such as going into another relationship, having a support system of friends, going out on weekends

to distract oneself and doing something new. In addition to romantic relationship breakups, there were references to friendship and family losses, which made participants feel lonely. These losses included someone close dying and moving away.

b) *Transitionary periods.* Transitionary periods such as moving from secondary school to university and from university to the workforce were another cause of loneliness. Young adults said they felt lonely during their transitions because the environment and responsibilities different drastically from what they were used to. For example, one goes from being used to attending lectures, making friends, and having the freedom to control one's day as a student to working from 9am-5pm with little flexibility. There was also mention of the shift in terms of working with people who are much older, hold different ranks and have been in the company significantly longer than oneself. Similarly, with regards to the transition from secondary school to university, participants said the new lifestyle and fresh set of responsibilities such as cooking, cleaning, budgeting, making friends and working part-time were overwhelming. In both contexts, participants said they felt lonely during the transition because they had no one to talk to or found it difficult to communicate with others while they saw that everyone was settled in, knew what they were doing and had friendship groups. They also mentioned that no one had told them about these transitions, the challenges associated with them and how they should prepare themselves:

*"it's entering a new place, for example, going to university or work I felt the most lonely. It's all about that transition phase, no one really talks about that big jump, when you go from university to work, when you go from college to university, it's like a completely different jump. It's a completely different level of work. Nobody talks to you about that transition, how you are going to fit in etc, and I felt really lonely because obviously I had to make completely new friends . . ."* [Female, 22, BAME, Tower Hamlets]

The consequences of romantic relationship breakups are symbolised as being more harmful than being single and lonely without a partner:

*"[Break ups] can make that lonely more worse. Yeah, it's like the opening from an old wound, it is gonna just unleash more bad stuff or negative stuff in that person's life."* [Male, aged 23, BAME, Tower Hamlets]

## Discussion

The present study aimed to explore the subjective causes of loneliness and the role played by social media in causing or mitigating loneliness among young adults from London's most deprived boroughs. The findings revealed that participants often feel they do not matter, are not understood or are unable to express themselves to others because they fear people will judge them negatively. They are also under considerable pressure to be accepted and social media play a large role in pushing them in the direction of focusing on increasing their social networks and how they appear to others in order to fit in. However, this is not felt to bring them wellbeing. Therefore, this study casts light on the equivocal findings on the impact of social media on loneliness in the literature and provides a triangulation of the quantitative studies. Since they can view other young adults' profiles and are exposed to their "highlights", even though they see social media as 'fake' representations of reality, young adults tend to compare themselves with these 'fake' representations and are prone to feel a sense of deficit. Consequently, they feel pressured to keep up, and as a result, present a version of themselves

on social media that is not reflective of who they truly feel themselves to be or how they feel in order to be included and accepted by others.

This suggests that there is a strong emphasis on appearance, neglecting what lies beyond the surface—an authentic sense of self. Hence, while increasing one's social networks may alleviate social loneliness in young adults, it leaves little space for tackling the emotional component of loneliness, which according to Weiss' theory of multidimensional loneliness is the more intensely painful form because one feels a lack of close and meaningful relationships [19]. Transitory educational, household and employment phases and the pressure associated with excessive work were also conceptualised as causes of loneliness.

Responses were compared across gender, race, and religion. Overall, there was strong homogeneity across the sample, however, young women were more likely to engage in social comparison than young men were. This is in line with existing literature (e.g., [64]). In particular, females across the BAME sample and those identified as having no religion were more likely than other demographic groups to socially compare themselves. Future research could examine this trend more closely.

Couched in a phenomenological epistemological framework with an inductive approach, the study used a novel free association technique, which enabled access to the deep-laid thoughts, feeling, emotions and non-conscious links made by young adults regarding their loneliness. As a result, the study was able to provide a rich exploration of the subjectivity of young adults' loneliness experience and their perspectives on the causes of their loneliness. Phenomenological analysis revealed that young adults share very similar experiences of loneliness and of what causes them to feel lonely. Regarding cause, the sense of emotional isolation, pressure, social comparison and experience of social media provide noted sources of loneliness for many, as do the transitions in education, relationships and employment typical of young adulthood. These corroborate and extend what previous research found [29, 30].

The current finding pertaining to transitory stages was also in line with evolutionary theory [26] since they call for social reorientation. Loneliness experienced during these transitions signals deficits in sufficient social relationships and encourages young adults to fix these deficits. This process of re-establishing a social network in a new environment increases chances of survival and progression in young adults.

The study's sample consisted of young adults from deprived areas, hence, the themes that emerged may be specific to them only. Having identified the causes of loneliness in deprived areas, a study investigating a matched sample from wealthier boroughs would further identify the particular causes of loneliness among those from non-deprived backgrounds. Whether they differ is an empirical question. Furthermore, one might want to make this into a mixed-method study where the quantitative aspect would include a sample with sufficient power to reliably identify whether there are differences in the causes of loneliness between those who live in more deprived versus wealthier boroughs.

## Practical implications

Interventions to tackle loneliness should focus on helping young adults feel heard, listened to, and understood. They should provide opportunities for self-expression without the fear of being judged. At the same time, young adults should be made aware of the negative consequences of social media use for loneliness and mental health, regarding the pressure it can put on them regarding acceptance, physical appearance, and popularity. These messages can be taught in educational institutions or take the form of videos disseminated on social media. These videos might also include suggestions for how to engage with social media platforms in ways that reduce loneliness: active communication by chatting with friends and limiting

passive consumption. This is consistent with the stimulation hypothesis, wherein social media are regarded as having the potential to stimulate the quality of existing relationships by extending them to online domains and to generate opportunities to create new friendships, thereby reducing loneliness [33, 65].

Furthermore, social media companies and interventions should encourage more authentic self-expression online since authenticity on social media is associated with greater wellbeing [66]. They should also promote the practice of gratitude to minimise unhelpful social comparison because this would allow young adults to focus on what they have versus what they see others as having. Journaling gratitude on a regular basis has been shown to improve wellbeing [67, 68].

In addition, given that many people use smartphones and social media as a way to distract themselves when using public transport, mindfulness, a theory-based intervention, may increase their attention to the present moment and foster face-to-face social interactions. Mindfulness has been shown to be a strong predictor of wellbeing [69], increase empathy [70] and reduce the amount of time people spend on their smartphones each day in both correlational and experimental studies [37, 71, 72].

Finally, with regards to transitory phases such as starting university or work, educational institutions and professional organisations should find ways to inform upcoming candidates about the potential challenges that they are likely to face when they begin their studies or employment so they can better prepare themselves. Increasing social capital, such as by forming close social relationships, is associated with lower levels of loneliness during the transition into university [73].

Young adults are a particularly lonely demographic in Western countries, yet little is known about the root causes of their loneliness. This study was the first systematic, in-depth analysis of the subjective causes of loneliness in this group. Young adults from London's most deprived communities often experience a sense of disconnection because they feel they do not matter, are not understood and are unable to express themselves. These feelings may stem from the significant pressure they experience in relation to expectations of contemporary social and working life. Social media increase these pressures because young adults view other people's profiles and socially compare themselves with them, which often results in negative feelings. Young adulthood is bound to many transitions, which can be challenging. Since loneliness is linked with mental health challenges such as depression and anxiety, understanding the root causes of loneliness in young adults will help intervention designers and policy makers tackle the issue before it progresses into more severe mental health challenges.

## Notes

1. A complementary study of the same dataset examined the experience of loneliness in young adults [38].

2. This is a social grade classification in the UK based on occupation representing those lower on social and economic status. It is used by the UK's Office for National Statistics among other organisations [74].

3. We eventually excluded the following three criteria from the Office for National Statistics study due to difficulty in recruitment: 'having little trust or a sense of belonging to one's neighbourhood', 'relationship status' (i.e., being single) and 'living arrangement' (i.e., living in a single-person household). We also did not recruit participants below the age of 18 due to ethical issues.

4. In this paper, the focus is on free associations and their elaborations. This is the phenomenological aspect of the study.

5. There were two parts to the study and the focus in this paper is on the first part.

6. The calculation was made on the basis of whether participants talked about a particular theme or not. When 30 percent more of a group being compared to another (e.g., males versus females) mentioned a theme, this was regarded as a noteworthy difference.

## Acknowledgments

Thanks to Dr Gemma Moore for her support in this project.

## Author Contributions

**Conceptualization:** Sam Fardghassemi, Hélène Joffe.

**Formal analysis:** Sam Fardghassemi, Hélène Joffe.

**Funding acquisition:** Sam Fardghassemi, Hélène Joffe.

**Investigation:** Sam Fardghassemi, Hélène Joffe.

**Methodology:** Sam Fardghassemi, Hélène Joffe.

**Project administration:** Sam Fardghassemi, Hélène Joffe.

**Supervision:** Sam Fardghassemi, Hélène Joffe.

**Writing – original draft:** Sam Fardghassemi, Hélène Joffe.

**Writing – review & editing:** Sam Fardghassemi, Hélène Joffe.

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
