## [Decision Letter · Decision Letter 0]

2 Nov 2021

PONE-D-21-26267The causes of loneliness: The perspective of young adults in London’s most deprived areasPLOS ONE

Dear Dr. Fardghassemi,

Thank you for submitting your manuscript to PLOS ONE. After careful consideration, we feel that it has merit but does not fully meet PLOS ONE’s publication criteria as it currently stands. Therefore, we invite you to submit a revised version of the manuscript that addresses the points raised during the review process.

We look forward to receiving your revised manuscript.

Kind regards,

Johnson Chun-Sing Cheung, D.S.W.

Academic Editor

PLOS ONE

Journal Requirements:

We would like to thank the UCL Grand Challenges Environment and Wellbeing initiative for funding this research.

SF

GM

156425 Grand Challenges

UCL Grand Challenges Environment and Wellbeing initiative 

https://www.ucl.ac.uk/grand-challenges/

This work was supported by a grant from the UCL Grand Challenges Environment and Wellbeing initiative (156425). The funders had no role in study design, data collection and analysis, decision to publish, or preparation of the manuscript. 

5. Please note that in order to use the direct billing option the corresponding author must be affiliated with the chosen institute. Please either amend your manuscript to change the affiliation or corresponding author, or email us at plosone@plos.org with a request to remove this option.

**Comments to the Author**

Reviewer #1: I am very thankful to you for allowing me to review the paper. The paper titled "The causes of loneliness: The perspective of young adults in London’s most deprived areas" is very well written paper, but it is qualitative. There is not much use of Statistics in the paper. It would be better to define a technique for sampling. The authors did not mention about the technique to identify the deprived areas of London. There is need to enhance the sample size of the study as well. Qualitatively, authors proved the results with regard to hypothesis.

Reviewer #2: The present manuscript aims to explore the subjective causes of loneliness among young adults who live in the most deprived area in London. This is a well-written manuscript and I enjoyed much reading it. Some comments that might further improve are listed below:

1. Line 194-195: The first research question is a general one (the subjective causes). The authors added the second question (the role of social media). When this is written as one of the aim, I would expect that the role of social media will be explored via a specific question during the interview. When it was not the case, authors mighty consider to exclude this research question.

2. Table 1: For consistency, also add percentage for each boroughs in the 'Total' column

3. Line 253 - 265 - Data Analysis: Please elaborate the steps of data analysis being done: how many coders work independently? When was the second coder did the double coded to ensure reliability? Any improvement or changes being done after this comparison? etc

4. Line 622-626: It is stated that responses were compared across gender, race and religion. Authors mentioned the females, BAME and no religion more likely to socially compare themselves. However, I could not find description in the Results section (theme 4: social comparison) regarding comparison between race and religions, only between gender. Please elaborate.

Line 646-647 - Limitation: I would argue that this is not a limitation, since the research question and aim of the present study was to explore loneliness in the young adults from deprived area, no comparison was intended.

---

## [Author Response · Author response to Decision Letter 0]

17 Dec 2021

We believe we have addressed all of the points raised by the editor and reviewers. Please see the attached documents. Many thanks.

---

## [Decision Letter · Decision Letter 1]

15 Feb 2022

The causes of loneliness: The perspective of young adults in London’s most deprived areas

PONE-D-21-26267R1

Dear Dr. Fardghassemi,

We’re pleased to inform you that your manuscript has been judged scientifically suitable for publication and will be formally accepted for publication once it meets all outstanding technical requirements.

Kind regards,

Miquel Vall-llosera Camps

Senior Editor

PLOS ONE

Reviewers' comments:

Reviewer's Responses to Questions

**Comments to the Author**

1. If the authors have adequately addressed your comments raised in a previous round of review and you feel that this manuscript is now acceptable for publication, you may indicate that here to bypass the “Comments to the Author” section, enter your conflict of interest statement in the “Confidential to Editor” section, and submit your "Accept" recommendation.

Reviewer #1: All comments have been addressed

Reviewer #2: All comments have been addressed

2. Is the manuscript technically sound, and do the data support the conclusions?

Reviewer #1: Yes

Reviewer #2: Yes

3. Has the statistical analysis been performed appropriately and rigorously? 

Reviewer #1: Yes

Reviewer #2: Yes

4. Have the authors made all data underlying the findings in their manuscript fully available?

Reviewer #1: Yes

Reviewer #2: Yes

5. Is the manuscript presented in an intelligible fashion and written in standard English?

Reviewer #1: Yes

Reviewer #2: Yes

6. Review Comments to the Author

Reviewer #1: Accepted. God Bless you.

Hope you will be able to write more such papers........................................................................................................................................................................................

Reviewer #2: (No Response)

7. PLOS authors have the option to publish the peer review history of their article (what does this mean?). If published, this will include your full peer review and any attached files.

Reviewer #1: No

Reviewer #2: No

---

## [Editor Report · Acceptance letter]

15 Mar 2022

PONE-D-21-26267R1 

The causes of loneliness: The perspective of young adults in London’s most deprived areas 

Dear Dr. Fardghassemi:

I'm pleased to inform you that your manuscript has been deemed suitable for publication in PLOS ONE. Congratulations! Your manuscript is now with our production department. 

Kind regards, 

on behalf of

Dr. Miquel Vall-llosera Camps 

Staff Editor

PLOS ONE